# Molecular Mechanisms Underlying the Association between Endometriosis and Ectopic Pregnancy

**DOI:** 10.3390/ijms23073490

**Published:** 2022-03-23

**Authors:** Julia Załęcka, Katarzyna Pankiewicz, Tadeusz Issat, Piotr Laudański

**Affiliations:** 11st Department of Obstetrics and Gynecology, Medical University of Warsaw, Starynkiewicza 1/3, 02-015 Warsaw, Poland; julia.zalecka@gmail.com; 2Department of Obstetrics and Gynecology, Institute of Mother and Child in Warsaw, Kasprzaka 17a, 01-211 Warsaw, Poland; katarzynahak@wp.pl (K.P.); tadeusz.issat@imid.med.pl (T.I.); 3OVIklinika Infertility Center, Połczyńska 31, 01-377 Warsaw, Poland

**Keywords:** endometriosis, ectopic pregnancy, reproductive immunology, epigenetics

## Abstract

Endometriosis is a common inflammatory disease characterized by the presence of endometrial cells outside the uterine cavity. It is estimated that it affects 10% of women of reproductive age. Its pathogenesis covers a wide range of abnormalities, including adhesion, proliferation, and cell signaling disturbances. It is associated with a significant deterioration in quality of life as a result of chronic pelvic pain and may also lead to infertility. One of the most serious complications of endometriosis is an ectopic pregnancy (EP). Currently, the exact mechanism explaining this phenomenon is unknown; therefore, there are no effective methods of prevention. It is assumed that the pathogenesis of EP is influenced by abnormalities in the contraction of the fallopian tube muscles, the mobility of the cilia, and in the fallopian microenvironment. Endometriosis can disrupt function on all three levels and thus contribute to the implantation of the embryo beyond the physiological site. This review takes into account aspects of the molecular mechanisms involved in the pathophysiology of endometriosis and EP, with particular emphasis on the similarities between them.

## 1. Introduction

Endometriosis is a common inflammatory disease characterized by the presence of endometrial and stromal gland cells outside of their physiological location [1]. It is estimated that symptomatic endometriosis affects 10% of women [2]. It seems that the most likely mechanism for the appearance of endometrial cells outside the uterine cavity is retrograde menstruation, proposed by Sampson [3]. Nevertheless, retrograde menstrual flow is a common event and there are other concomitant factors necessary for the development of the disease. The source of endometrial ectopic cells may be mesothelium, stem cells, celomic metaplasia, bone marrow stem cells, and vascular or lymphatic dissemination [4]. Neither of the theories fully explains the complexity of the pathogenesis of endometriosis; therefore, it is postulated that various genetic, epigenetic, and environmental factors contribute to the elicitation of the disease [5]. There is a wide range of abnormalities observed in endometriosis, including adhesion, proliferation, and cell signaling disturbances. The development of ectopic endometrial tissue is related to endocrine, pro-angiogenic, immunological, and inflammatory processes, with the production of a number of cytokines, chemokines, and prostaglandins [6]. Endometriosis is associated with reduced fertility in women of childbearing age and with chronic pelvic pain, which significantly impacts quality of life [4].

Ectopic pregnancy (EP) is a serious complication of early gestation and is defined as a pregnancy implanted outside the uterine cavity. It affects 1–2% of all pregnant women and is the leading cause of maternal death in the first trimester [7]. Most often, incorrect implantation takes place in the fallopian tube (tubal ectopic pregnancy TEP) due to the physiological passage of the embryo [8]. The mechanism leading to an EP is not fully understood. Most of the data come from in vitro studies as ectopic pregnancy is rare in animals and it is difficult to develop such a model [9]. However, it can be assumed that the passage of the oocyte and later the embryo along the fallopian tube is influenced by three main components: ciliary movement, contractility of the fallopian tubes, and tubal fluid. All of these phenomena occur due to the paracrine relationship between the epithelium and endothelium of the fallopian tube, the immune system, and the embryo [10]. 

It is not only clinical observation that might suggest that ectopic pregnancy may be associated with endometriosis due to altered tubo-ovarian anatomy, but also several cohort and case-control studies have shown such links. The most recent metanalysis of 15 studies presented that there exist possible association (OR = 2.16–2.66), and continued research is needed to delineate the pregnancy implications of endometriosis [11]. Further, published in the same year, analysis of the Nurses Health Study II confirmed an association between endometriosis and ectopic pregnancy [12]. Since there are several possible etiological mechanisms for the association between endometriosis and ectopic pregnancy, we decided in this narrative review to summarize the knowledge about the pathophysiology of both endometriosis and EP with particular attention on possible relationships between their occurrence.

As mentioned above, endometriosis is a known risk factor for EP. Although to date there are no studies proving that early identification or certain management of the one can prevent the onset of the other, finding the common molecular mechanisms of both diseases might be of high importance from a clinical point of view. Women with endometriosis are often treated because of infertility with assisted reproduction techniques (ART), whereas the use of ART alone is one of the most relevant risk factors for EP, putting these patients at a very high risk of developing EP. Further research is needed to explore different possibilities of EP prevention, especially in patients with endometriosis. This review may be a starting point for planning studies aiming to find potential molecular targets for the prevention and therapy of both EP and endometriosis. 

## 2. The Potential Keypoints in the Association between EP and Endometriosis

Decidualization is a physiological process by which the normal endometrium prepares itself to receive the embryo properly under the influence of pregnancy hormones. A similar hormonal response can also be observed in the case of the ectopic endometrium. [13]. Thus, cells outside the uterus may also predispose the embryo to implantation. Otherwise it would seem that endometriosis can affect the frequency of cilia beats, mechanically block the descending oocyte or embryo, and create an environment that promotes implantation and development beyond the endometrium [14]. Prostaglandins and progesterone are proposed modulators of ciliary movement [15]. When looking at the physiological work of cilia, adrenomedullin (ADM) deserves special attention. It is a protein that is steroid modified and highly expressed in the epithelium of the fallopian tube. Decreased levels of ADM and its messenger RNA (mRNA) have been observed in women with TEP. It has been suggested that ADM is responsible for the proper transport of the embryo in the fallopian tube. In addition, it is known that ADM has a strong anti-inflammatory effect; therefore, reduced ADM expression may be responsible for the occurrence of an excessive immune response favoring TEP [16]. 

For the embryo to establish itself in the wall of the fallopian tube, a network of vessels is needed for further development. Additional vascularization is associated with increased expression of angiogenesis-promoting factors [17]. It has been shown that the level of vascular endothelial growth factor (VEGF) is elevated at the site of embryo implantation in the fallopian tube compared to other sections, and its concentration in the serum is increased compared to women without TEP [18]. 

During the first trimester of pregnancy, extravillous trophoblast cells (EVT) invade the maternal decidua. Invasion normally is reduced from the second trimester onwards and stops in the inner third of the myometrium. In TEP, due to specific immunological microenvironments, apoptosis induction fails, which deleteriously may result in uncontrolled invasion and penetration of the tubal wall. This process may be enhanced by endometriotic lesions through the Slit2/Robo1 signaling pathway [19,20]. It has been observed that in normal pregnancy, the endometrium becomes receptive as a result of pro-inflammatory cytokine interactions. Thus, in the pathophysiology of TEP, the inflammatory environment and intermolecular signaling induce tubal receptivity, increasing embryo adhesion and invasion, leading to implantation [21]. It is also worth mentioning the possible role of hormones in the pathogenesis of TEP: 17-ß-estradiol is responsible for many physiological processes in the fallopian tube. Therefore, its disturbances can result in the formation of different abnormalities of the fallopian tube, including TEP. However, it is not known whether endocrine disruption is the cause of TEP or just an accompanying symptom [22]. 

In recent years, researchers have started to consider the cannabinoid system and its influence not only within the nervous system but also observed some pathological patterns in the reproductive system, which may also have implications for TEP. In the further part of the review, each of the above-mentioned issues has been developed and presented in Figure 1.

## 3. Inflammatory Environment

The immune system has a strong influence on the pathogenesis of endometriosis, its course, and symptoms. The peripheral inflammatory and local responses are impaired in women with endometriosis. They are considered a possible cause of disease progression and related pain. As a result of endometriosis, reproductive function can be impaired, consequently leading to infertility. This is probably due to disturbed inflammatory processes that affect the oocyte and further development of the embryo [23]. The cells that play a key role in endometriosis are macrophages. Owing to their ability to produce and activate cytokines, both inflammatory and anti-inflammatory, they can participate in the formation of ectopic lesions and are responsible for the unfavorable environment for the development of pregnancy [24]. In a study by Mei et al., it was shown that macrophages cultured with ectopic stromal cells of the endometrium (ECS) displayed reduced phagocytic activity. In turn, induced macrophages significantly increased the proliferation and viability of the ECS [25]. This suggests a significant role for the cross-signaling between the ECS and macrophages, but the exact mechanisms are not yet well understood. Ahn et al. provided evidence that endometrial lesions produce IL-17A, and their laparoscopic removal significantly lowered the systemic level of IL-17A [26]. Looking for the role of this cytokine in the development of endometriosis, Miller et al. observed that IL-17A treatment of the endometriotic epithelial cell line produced macrophage activating and recruiting cytokines including G-CSF, GM-CSF, CXCL-1, and IL-8 [27]. Moreover, researchers suggest that IL-17A may mediate neutrophil accumulation in the site of inflammation by promoting G-CSF [28]. 

It is believed that the imbalance between the phenotypes of M1 and M2 macrophages with M2 dominance may contribute to the escalation of the disease by modulating the immune response [29]. However, this division is not always reflected in the complexity of the macrophage phenotype in which pro-inflammatory and homeostatic markers often coexist depending on signals from the local microenvironment [30]. The proliferation of lesions is also influenced by VEGF produced by macrophages, which strongly promotes angiogenesis [31]. Dendritic cells, similar to macrophages, may increase angiogenesis and thus support the development of ectopic lesions [32]. Neutrophils have also been found to influence VEGF production in the course of endometriosis. These cells contribute to the growth and survival of cells outside the uterine cavity. Studies have reported an increase in neutrophils in the peritoneal fluid and their pro-inflammatory cytokines such as IL-8, CXCL10, and reactive oxygen species [33]. An increased concentration of IL-17 produced by mast cells has also been demonstrated, which might contribute to the progression of inflammation [34]. Analysis of peritoneal fluid in women with endometriosis showed increased levels of pro-inflammatory and pro-angiogenic cytokines IL-6 and IL-8, which is consistent with previous observations [35]. 

Endometrial ectopic cells can be implanted not only in the pelvic wall but also in the fallopian tube. It is estimated that this is about 60% of the cases of women with endometriosis [36]. It has been demonstrated that in women with tubal endometriosis, the muscle activity of the fallopian tube and cilia is impaired, which may contribute to infertility [37]. The same mechanism occurs in women with a tubal pregnancy. In tubal endometriosis, ectopic changes have been observed mainly in inflammatory pathways and acute phase responses. As in the peritoneal fluid, elevated levels of IL-6, TNF-alpha, C2, and C4B have been demonstrated [38]. It is possible that IL-6 is responsible for the growth and further development of lesions [39] and its expression, as a result of downregulation of dual-specificity phosphatase-2 (DUSP2), allows it to survive in the ectopic environment [40]. 

In the study by Goryszewska-Szczurek et al., trophoblast expression of PROK1 mRNA increased gradually and acted through PROKR1 to regulate the expression of pregnancy-related genes. Moreover, it increased trophoblast proliferation via PI3K/AKT/mTOR, MAPK, and cAMP signaling pathways and adhesion via MAPK and PI3K/AKT signaling pathways [41]. Moreover, Wu et al. observed that TNF-alpha promoted angiogenesis by increasing PROK1 expression [42]. On the other hand, by looking at the tissues of the fallopian tube during ectopic pregnancy, the positive correlation between PROK receptors and inflammatory cytokines genes was investigated. Increased expression of PROKR1 and PROKR2 was demonstrated at the site of embryo implantation. In addition, TNF-alpha, IL-8, and IL-6 concentrations were increased at the implantation site. The pro-inflammatory environment may influence blastocyst adhesion and invasion. Moreover, it has been proposed that dysregulation of PROKR2 and IL-8 may be risk factors for the occurrence of TEP [43]. Earlier studies also indicated an increased level of IL-8 and IL-6 at the site of implantation [44], and their concentrations may result from the downregulation of ADM, which has anti-inflammatory properties and is responsible for the ciliary movement in the fallopian tube [16]. 

Inflammation induction can also be caused by B-cell activation factor (BAFF), which regulates acquired and innate immune responses. Increased BAFF expression with simultaneous increases in IL-6 and TNF-alpha levels was observed in patients with TEP [45]. In addition, Hever et al. confirmed that BAFF was elevated in the endometriotic lesions and in the serum of the patients with endometriosis compared to the control [46]. Based on the above results, it can be concluded that inflammatory factors have a significant influence on the pathogenesis of TEP, which may be the reason for the more frequent occurrence of ectopic pregnancy in women with endometriosis. Lekovich et al. observed a correlation between elevated serum levels of interleukin-1β and more frequent occurrence of EP [47]. In turn, Shi et al. showed that higher concentrations of IL-1β in patients with endometriosis led to an increase in WEE1 protein kinase, which is responsible for the induction of fibrosis. Moreover, they reported that WEE1 overexpression has been associated with a significant increase in ß-catenin, hence the supposition that abnormal inflammatory patterns may affect the expression of adhesion molecules via the Wnt/ß-catenin pathway [48].

The other important possibility is that ectopic pregnancy occurs because of conflicting signals to the blastocyst from the uterine and fallopian tube epithelium. The signals consist of cytokines, chemokines, and adhesion molecules that mediate both blastocyst adhesion to the uterine (and fallopian) epithelium and leukocyte adhesion to the vascular endothelium and, presumably, the fallopian epithelium. Chronic inflammation in the fallopian tube caused by infections can also alter expressions of the signals sent from the fallopian tube and thus compete with the uterine site of implantation. This means that blastocyst may receive stronger signals from the tubal epithelia, migrate to the fallopian tube, and be implanted at that site [49]. Blastocyst implantation is an integrin-dependent process that occurs in a chemokine–cytokine-rich microenvironment, which is formed by the epithelium and decidua. It is likely that impaired tubal epithelia secrete many inflammatory cytokines and chemokines that may promote and direct the blastocyst to incorrectly migrate to the wrong foci during implantation. These factors include i.e., leukemia inhibitory factor (LIF), TNFα, TGFβ, IL-1 and IL-8, L-selectin, and throphinin [50]. They are often related to pelvic inflammatory disease, especially with *Chlamydia trachomatis* infection. It is estimated that acute salpingitis increases the risk of developing EP 7-fold. Abnormal signaling from Fallopian tubes may be also related to scar tissue formed from a previous cesarean section or endometrial surgery, as well as to endometriosis, where persistent inflammatory response-type action becomes an attractive embryo implantation site [49].

## 4. The Wnt/ß-Catenin Signaling Pathway

Wnt family signaling molecules are important in epithelial–mesenchymal cell interactions and have unique expression patterns in various types of endometrial cells [51]. Inflammation can alter the intermolecular signals of the fallopian tube tissues and predispose women to conditions for ectopic implantation. In rodent studies, it was shown that Wnt expression influenced the decidualization and development of the embryo, but also the retention of the blastocyst in the fallopian tube [52,53]. In the Wnt signaling pathway, the major effector role is played by ß-catenin, which binds to E-cadherin in the epithelium [54]. Li et al. observed increased expression of ß-catenin in secretory and ciliated cells of the fallopian tube epithelium, with its particular increase at the site of embryo implantation. This may suggest its special role in TEP. Additionally, an increase in the concentration of ß-catenin in fallopian tubes with chronic inflammation was demonstrated. The results correlated with a reduced concentration of E-cadherin in the fallopian tube tissue, which may be responsible for the invasion of the trophoblast into the tubal epithelium. Moreover, they observed tubal artery hyperplasia and an increase in glycogen accumulation in tubal tissue cells in patients with TEP, which is consistent with the effects induced by Wnt signaling and inflammation [55]. All of these changes can create a favorable environment for the ectopic implantation of embryos. 

Matsuzaki et al. suggested that inhibition of active MMP2 and MMP9 by treatment with PKF 115–584, an antagonist of the Tcf/ß-catenin complex, decreased the numbers of invasive endometriotic epithelial cells and stromal cells [56]. Moreover, treatment with Wnt3a induced clearly visible α-SMA-positive stress fibers in endometrial stromal cells of patients without endometriosis, suggesting the involvement of the Wnt/ß-catenin pathway in the fibrogenesis process [57]. It has been shown that Wnt/ß-catenin signaling can facilitate the growth and development of ectopic changes through increased cell migration, invasion, and proliferation capacity. Abnormalities in molecular signaling may contribute to the progression of accompanying fibrosis in the course of endometriosis [58]. Additionally, Zhang et al. suggested that the Wnt/ß-catenin pathway could be induced by 17-ß-estradiol (E2) and contribute directly to the expression of *VEGF* genes [59]. Moreover, in a mouse model, it was shown that disruption of estrogen signaling in fallopian epithelial cells affects transcripts in the Wnt/ß-catenin pathway, thus modifying cilia function and impairing embryo transport [60]. This information indicates an important role of hormonal regulation in the pathophysiology of both endometriosis and EP. Potential relationships of Wnt/ß-catenin pathway, hormonal regulation and inflammation are presented in Figure 2.

## 5. The Role of Hormones

Hormonal balance, especially E2 with its receptors ESR1 and ESR2, plays a key role in the physiological processes in the reproductive system. Estrogen and progesterone secretion is crucial in hormonal regulation of both EP and endometriosis.

The main cell types of the fallopian tube are ciliated and secretory epithelial cells, smooth muscle cells, immunocompetent cells such as leukocytes, and blood vessel cells. In normal human fallopian tubes, both ERα and ERβ are coexpressed at similar levels. However, these two ER subtypes are regulated by different mechanisms. During the menstrual cycle, the levels of ERβ expression fluctuate in response to high circulating E2 levels whereas the levels of ERα expression are not altered. ERα is frequently lost in the implantation and non-implantation site of the Fallopian tube in women who have suffered from ectopic pregnancy [22]. 

Estrogens determine ADM levels in the fallopian tube, which contribute to the frequency of ciliary movement and smooth muscle contractility. Disturbances in E2 concentrations may result in decreased ADM expression and lead to the impaired function of the fallopian tube [61]. However, in a mouse model, it was shown that blastocyst retention in the fallopian tube did not result in TEP, hence the conclusion that there must be other mechanisms to promote implantation outside the uterine cavity [62]. Li et al. observed differential expression of cytokines in TEP fallopian tubes compared to the control. Researchers treated tubal TEP samples with ADM, which significantly lowered the levels of previously elevated pro-inflammatory cytokines: Granulocyte-macrophage colony-stimulating factor (GM-CSF), IL-7, TNF-beta; anti-inflammatory: hepatocyte growth factor (HGF), IL-10; as well as those of a mixed nature: IL-6, IL-4 [63]. This suggests that estrogens may influence ectopic implantation through ADM failure. Li et al. demonstrated increased expression of the S100 calcium-binding protein A (S100A) genes in E2-treated sheep oviduct epithelial cells (SOECs). The levels of the inflammatory factors, interleukin IL -1β, and IL-4 were significantly increased in the S100A8 knockout SOEC, while the levels of the anti-inflammatory factor IL-10 were decreased. Moreover, E2 compensated for the suppression of S100A8 by increasing IL-10 [64]. Barton et al. reported a positive effect of estrogen on the secretory functions in the fallopian tube, as well as the formation of the tubal protein, which is essential for the proper development of the embryo. For this reason, they emphasized that disruption of estrogen signaling may favor the occurrence of TEP [65]. 

As in the case of the fallopian tubes, estrogens and their receptors play an essential role in the survival of endometrial tissues. Increased expression of ESR2, related to hypomethylation of its promoter, has been reported in ectopic lesions in the course of endometriosis. It has been described that this overexpression disrupts signaling for TNF-alpha-induced apoptosis [66]. Additionally, it is involved in the processes of detoxification of oxygen radicals and in signaling pathways related to hypoxia. ESR2 appears to be able to protect ectopic cells from an immune response, as well as increase their potential for invasion and progression [67]. 

Zhang et al. suggest that endometriotic lesions undergo repeated tissue damage and repair by activating the TGF-β1/Smad3 signaling pathway. As a consequence, they undergo epithelial-mesenchymal transition (EMT) and transdifferentiation of fibroblasts into myofibroblasts (FMT), causing increased smooth muscle metaplasia (SMM) and collagen production, leading to fibrosis [68]. ACTA2, also known as alpha smooth muscle actin (α-SMA), is the most widely used marker of myofibroblasts, increasing their contractile activity [69]. Xu et al. investigated the role of ACTA2 in promoting migration and invasion of eutopic endometrial stromal cells (euESCs). They showed that ACTA2 expression was regulated by H19 by competition for inhibitory miR-216a-5β binding sites, and reducing these inhibited euESC invasion and migration. Moreover, as a result of the estrogen treatment, there was an increase in migration. The presented results indicate that changes in the estrogen/H19/miR-216a-5p/ACTA2 pathway may contribute to the recruitment of euESC, resulting in endometriotic lesions [70]. 

Horne et al. investigated the expression of estrogen, progesterone (PR-A, PR-B), and androgen (AR) receptors in the tissues of the fallopian tube during the menstrual cycle. There was only a significant difference in PR-AB and PR-B mRNA expression, i.e., it was significantly reduced in the luteal phase as compared to the follicular phase. Additionally, they showed that the occurrence of TEP correlated with decreased expression of PR-B mRNA compared to expression in the non-pregnant fallopian tube in the mid-luteal phase of the cycle [71]. Moreover, progesterone has been shown to rapidly reduce the frequency of cilia beats via PR, suggesting that aberrant progesterone signaling may contribute to TEP [72]. In endometriosis, both forms of the progesterone receptors PR-A and PR-B are decreased, especially PR-B expression through promoter hypermethylation [73]. As a consequence, endometriotic lesions acquire resistance to progesterone, which, according to Koninckx et al. may underlie the development of endometriosis [74]. Progesterone resistance is attributed to extremely low PR levels observed in vivo in this tissue. In normal endometrium, levels of the PR isoforms, PR-B and PR-A, progressively increase during the proliferative phase, peak immediately before ovulation, and diminish after ovulation, suggesting that estradiol stimulates PR levels. In contrast, PR-B is undetectable, and PR-A is markedly lower in vivo in simultaneously collected endometriotic tissues.

Noteworthy is also a potential role of other hormones in the pathophysiology of EP and endometriosis. Peng et al. localized the gonadotrophin-releasing hormone (GnRH) and its receptor (GnRHR) in tubal biopsies at the sites of pregnancy implantation. In addition, GnRH and GnRHR immunoreactivity was detected in cytotrophoblasts, syncytiotrophoblasts, and extracellular trophoblasts in all TEPs [75].

Dobovišek et al. suggested that estrogen signaling may be related to the action of cannabinoids in the treatment of ESR-expressing breast cancer [76]. So far, there are no clinical trials on the effects of cannabinoids in both endometriosis and EP, but their potential role in the pathogenesis of these diseases is worth summarizing.

## 6. The Endocannabinoid System

The endocannabinoid system (ECS) consists of cannabinoids and endocannabinoids, their receptors, and enzymes that regulate the synthesis and degradation of this system [77]. ECS influences many physiological and pathological processes. The most famous and studied endocannabinoid is N-arachidonoyl-ethanolamine, known as anandamide (AEA). The synthesis of AEA begins with the production of N-arachidonoyl phosphatidylethanol (NAPE) and can follow four routes involving NAPE- phospholipase C (PLC) and phosphatase, N-arachidonoyl phosphatidyl ethanol-preferring phospholipase D (NAPE-PLD), phospholipase B (ABHD-4), and lyso-PLD, ABHD4 and glycerophosphodiesterase (GDE-1). On the other hand, AEA inactivation occurs with the enzyme fatty acid aminohydrolase (FAAH) or, to a lesser extent, with oxidation by cyclooxygenase-2 (COX-2) [78]. Cannabinoids interact with their receptors, the most common of which are the cannabinoid CB1 and CB2 receptors. Their activity is coupled with the G protein, but they differ in the place of their occurrence. The main site of CB1 expression is the nervous system, while CB2 is dominant in the immune system [79]. Elements of the ECS are also involved in the functioning of the reproductive system, in both males and females [80]. 

CB1-mediated endocannabinoids are believed to inhibit the release of hypothalamic gonadotropin-releasing hormone (GnRH) and thus may impair reproductive function [81]. Additionally, an appropriate balance between AEA synthesis and degradation is important in such processes as follicle development, oocyte maturation, and ovulation [82]. It has been observed that the level of AEA in the uterus changes with the state of pregnancy [83]. Performed studies suggest that high serum levels of AEA are required during ovulation [84]; however, successful implantation itself correlates with lower AEA serum concentrations [84,85]. This is related to the endometrial decidualization process that takes place at low AEA concentrations [86]. Under physiological conditions, AEA levels fluctuate during the menstrual cycle. It was found that the lowest levels of AEA and the highest levels of FAAH coincide with the occurrence of the implantation window, the point at which the blastocyst can properly interact with the receptive endometrium [87]. It is possible that the high levels of AEA are due to a deficiency of cannabinoid-inactivating enzymes. The reduced activity of FAAH in peripheral lymphocytes is associated with an increased miscarriage rate, suggesting an adverse effect of high levels of endocannabinoids on pregnancy and embryonic development [88]. Additionally, it was confirmed in a mouse model that suppression of FAAH action in the fallopian tubes results in impaired transport through the fallopian tube, embryo retention, and incorrect implantation, which is consistent with previous observations [89].

Focusing on the passage of the embryo through the fallopian tube, studies were carried out on mouse models. The cannabinoid CB1 receptor was suppressed pharmacologically or genetically, resulting in the retention of numerous embryos in the fallopian tube. This process was reversible with a beta-adrenergic agonist. This study suggests that the coordinated action of cannabinoid and adrenergic signaling is involved in normal contractions and relaxation of the fallopian tube muscles, which are necessary for the proper transport of the embryo [90]. Moreover, the variability of CB1 expression in the human fallopian tube was demonstrated and it was higher in the luteal phase compared to the follicular phase. Available research show that endocannabinoids in EP can also affect the function of the fallopian tubes. In patients with EP, lower expression of CB1 mRNA in the fallopian tube and endometrium was observed compared to the group of women with intrauterine pregnancy [91]. 

To date, little is known about the role of ECS in the pathogenesis of endometriosis; however, cannabinoids may contribute to the inflammation closely related to endometriosis [92]. It has been demonstrated that M1 macrophages stimulated AEA production by granular cells, which may explain the increased level of AEA in the follicular fluid in the course of endometriosis [93]. Moreover, similarly to EP, increased levels of systemic AEA were shown along with decreased expression of CB1 in the secretory phase as compared to the control. In addition, patients with severe painful periods were characterized by a higher level of AEA compared to those with mild pain [94]. Earlier studies also obtained results with decreased CB1 expression in endometriosis, regardless of the phase of the cycle [95]. 

A recently published study indicated that phytocannabinoids may affect follicular ECS signaling and the epigenome in the surrounding granulosa cells [96]. Referring to epigenetic changes and their role in the pathophysiology of endometriosis and EP, we are obligated to discuss the role of microRNAs (miRNAs) in this process.

## 7. The Role of microRNA

MiRNAs are a class of noncoding RNAs, consisting of 21–22 nucleotides and formed through the activation of the Drosha and Dicer ribonuclease III endonucleases. They do not translate proteins, but can regulate gene expression by silencing mRNA translation and causing target mRNA degradation [97,98]. There is a vast body of evidence for a crucial role of miRNAs in the development and potential treatment of many diseases, including reproductive health diseases, such as premature ovarian insufficiency (POI), polycystic ovarian syndrome (PCOS) and infertility, as well as pregnancy complications (i.e., preeclampsia and fetal growth restriction) [99,100,101,102,103]. 

MiRNA expression profiles in the physiological and pathological endometrium are currently under intensive research. In our own studies, we also showed significant differences in eutopic endometrium of women with endometriosis as compared to controls [104,105]. It is well documented that miRNA expression in the human endometrium is dependent on the menstrual cycle phase and is regulated by sex steroids [106]. A broad range of different miRNAs involved in the pathogenesis of endometriosis and ectopic pregnancy has been described and they are summarized in Figure 3 [35,106,107,108]. In this part of the review, we focus on miRNAs that may constitute a link between these two diseases.

Dominguez et al. investigated the miRNA profile of embryonic tissues in ectopic pregnancies and controlled voluntary abortions. They found four miRNAs that were downregulated and three miRNAs that were upregulated in EP. Among the upregulated miRNAs were miR-223 and miR-451. Functional analysis demonstrated that the most significant pathways related to the abnormally expressed miRNAs were the mucin type O-glycan biosynthesis and the ECM-receptor-interaction pathways. Moreover, the dysregulation of three miRNAs (has-miR-196, hsa-miR-223, and hsa-miR-451) was able to alter the expression of the gene targets included in these pathways, such as the *GALNT13* and *ITGA2* genes [109]. On the contrary, Lu et al. presented that in women with EP serum, miR-223 and miR-873 concentrations were significantly lower than in women with viable intrauterine pregnancy [110]. In patients with endometriosis, miR-451 was found to be downregulated in the eutopic endometrium in comparison to the healthy controls, and it was probably related to the promotion of proliferation and inhibition of apoptosis of endometrial cells [111]. MiR-223 was also found among 14 upregulated miRNAs in eutopic and ectopic endometrial tissue in patients with endometriosis in the study performed by Ohlsson Teague and colleagues. Functional analysis suggested that these miRNAs were related to the pathways previously described in endometriosis, including c-Jun, CREB-binding protein, protein kinase B (Akt), and cyclin D1 (CCND1) signaling [112]. It is noteworthy that miR-223 was first described as a regulator of myelopoietic differentiation in 2003, and to date, its multiple regulatory functions in the immune response are well documented. Abnormal expression of miR-223 is associated with multiple inflammatory diseases, including myocarditis, type II diabetes, acute lung injury, rheumatoid arthritis, and inflammatory bowel disease, as well as infectious diseases, including viral hepatitis, human immunodeficiency virus type 1 (HIV-1), and tuberculosis [113]. 

Another important miRNA group involved in the development of both EP and endometriosis is the Lin28/Let-7 system. Lin28 is an RNA-binding protein able to bind to the precursors of Let7 miRNAs, thereby blocking their capacity to interact with Dicer [114]. Lozoya et al. performed a study evaluating the expression patterns of Lin28B mRNA and the related Let-7, miR-132, and miR-145 in human embryonic tissue from early normal and ectopic gestation. In normal intrauterine pregnancy, there was a significant increase in Lin28B mRNA expression with a significant decrease in Let7a, miR-132, and miR-145 expression after six weeks of pregnancy. On the contrary, in TEP, the expression of Lin28B was abnormally high, whereas Let-7a miRNA expression was already suppressed before six weeks of pregnancy and remained suppressed in the further course of pregnancy. The expression of miR-132 and miR-145 did not differ between normal and ectopic pregnancy. The authors of this study hypothesized that the Lin28/Let-7 system might be associated with the proliferative and invasive trophoblast phenotype and its dysregulation may lead to the disturbed trophoblast invasion commonly seen in EP [115]. Yi Feng et al. revealed that a significant decrease in DICER1 expression in patients with EP was followed by significant aberrations in the expression of multiple miRNAs including let-7i, miR-149, miR-182, and miR-424. NEDD4, TAF15, and SPEN genes have been recognized as targets for these miRNAs in functional analysis [116]. The let-7 family miRNAs dysregulation has been also demonstrated in endometriosis. Cho et al. found significantly decreased levels of circulating let-7b and miR-135a in women with in comparison to healthy controls. Moreover, let-7b expression strongly correlated with serum CA-125 levels [117]. Grechukhina et al. discovered inherited polymorphism of a let-7 miRNA binding site in KRAS in patients with severe endometriosis [118]. The role of let-7 miRNA is currently under intensive research as a potential biomarker and therapeutic target in endometriosis and gynecological cancers as well [119,120]. 

Romero-Ruiz et al. investigated the role of kisspeptins in EP. They found that circulating kisspeptins levels, as well as the expression of *KISS1* in embryonic/placental tissue, were significantly decreased in EP in comparison to voluntary terminated pregnancies. After bioinformatic analysis, miR-27b-3p and miR-324-3p were identified as potential repressors of *KISS1;* however, a significant repressive interaction with 3′-UTR of *KISS1* was confirmed only for miR-324-3p. Interestingly, although tissue overexpression of the pre-miRNA was detected, circulating levels of miR-324-3p were significantly decreased in women with EP [121]. Kisspeptins have essential roles in many biological processes necessary for normal pregnancy development, including endometrial receptivity and implantation via regulation of leukemia inhibitory factor (LIF), angiogenesis by VEGF, and trophoblast invasion via matrix metalloproteinases (MMPs) [122]. There is also evidence for the important role of kisspeptin 1 (KISS1) and its receptor (KISS1R) in the pathogenesis of endometriosis. Abdelkareem et al. studied the expression of KISS1 and KISS1R in eutopic and ectopic endometrial tissue of women with and without endometriosis. They demonstrated that KISS1 and KISS1R levels are downregulated in eutopic endometrial stroma from women with endometriosis versus those without the disease. Moreover, deeply invasive lesions showed lower KISS1 levels than superficial lesions, and the authors concluded that downregulation of KISS1 may be associated with implant invasiveness [123]. 

It is noteworthy that miR-324-3p is related not only to the kisspeptin molecular pathway but also the Wnt/ß-catenin signaling pathway, which is important in the pathophysiology of both EP and endometriosis. The association between miR-324-3p and WNT2B was confirmed in PCOS and prostate cancer [124,125]. 

There is a need for future studies aimed at finding miRNAs that regulate the expression of EP- and endometriosis-associated genes, such as VEGF, ESR1, IL-6, and IL-8. Expression analysis of these miRNAs can provide new insights into the mechanisms involved in EP- and endometriosis-pathogenesis and may contribute to develop new tailored therapies for such patients. 

## 8. Conclusions

This review provides an initial summary of the common features of the pathophysiology of endometriosis and EP. It is still unclear how strongly the two conditions are connected to each other and how. We presented studies that showed similarities for both pathologies. The fact is that both endometriosis and EP are associated with the inflammatory environment, the Wnt/ß-catenin pathway, and hormonal regulation that to date seems to be most important and well-documented common molecular mechanisms of both diseases. The potential associations of these factors are presented in Figure 2. Additionally, the ECS remains an interesting path for consideration, with similar changes observed in both pathological states. An abnormal miRNA profile may also contribute to the promotion of EP in endometriosis by regulating gene expression. It is possible that the presented relationship is the result of the overlapping of several factors included in this review, but also those unidentified. Further research is needed to improve knowledge about each of the potential pathways and to provide significant evidence for the link between endometriosis and EP. Understanding this relationship has the potential to prepare some prevention strategies for women with endometriosis to avoid EP occurrence. 

## Figures and Tables

**Figure 1 ijms-23-03490-f001:**
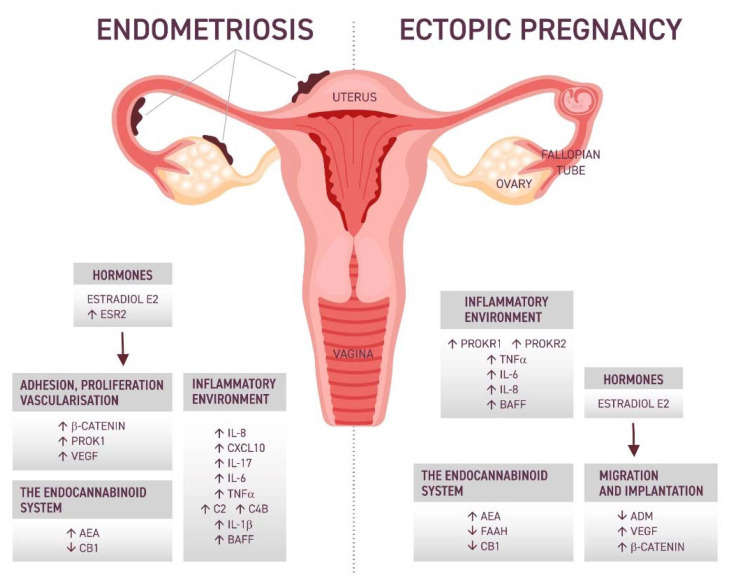
Comparison of potential factors involved in the pathophysiology of EP and endometriosis. ESR2-estrogen receptor, PROK1-prokineticin, PROKR1-, PROKR2-prokineticin receptors, VEGF-vascular endothelial growth factor, BAFF-B-cell activation factor, AEA-anandamide, FAAH-fatty acid aminohydrolase, CB1-cannabinoid receptor, ADM-adrenomedullin.

**Figure 2 ijms-23-03490-f002:**
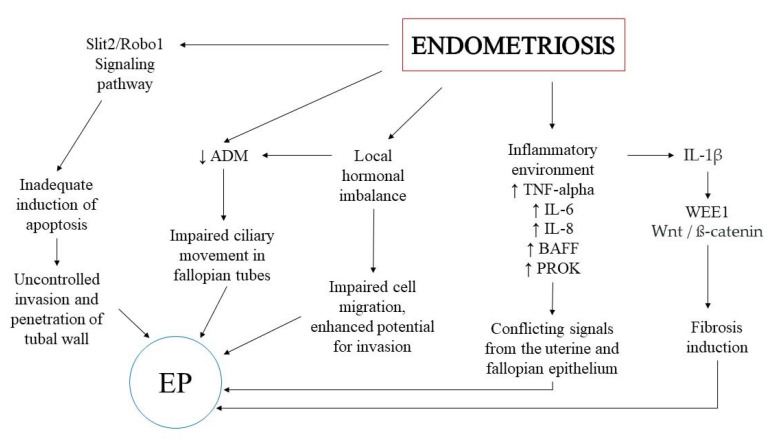
Potential relationships between endometriosis and EP related to the inflammatory environment, the Wnt/ß-catenin pathway, and hormonal regulation.

**Figure 3 ijms-23-03490-f003:**
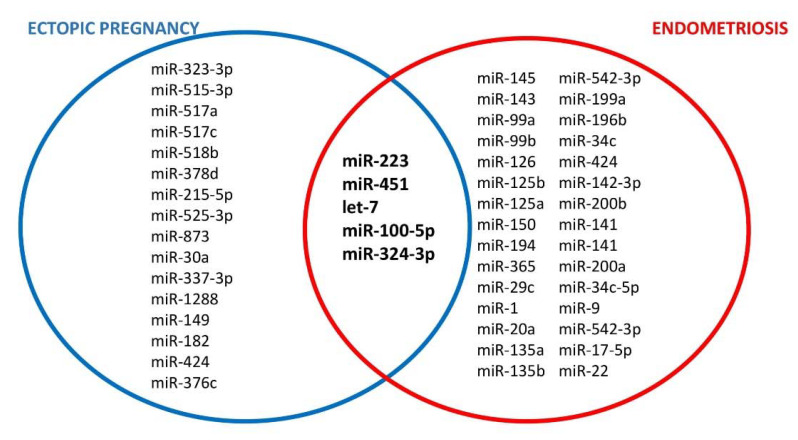
The most common miRNAs involved in the pathophysiology of EP and endometriosis.

## Data Availability

No new data were created or analyzed in this study. Data sharing is not applicable to this article.

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
