# Peer review of "Molecular Mechanisms Underlying the Association between Endometriosis and Ectopic Pregnancy"

_ijms, 2022, doi:10.3390/ijms23073490_

Round 1
Reviewer 1 Report
In this review, Załęcka and collaborators summarized the role of inflammation, WNT pathway and microRNAs in the pathogenesis of endometriosis and ectopic pregnancies.
The purpose of the review is important and relevant, however the authors were superficial in describing the proposed mechanisms and related roles of inflammation in endometriotic lesions and the mechanisms underlying ectopic pregnancy due to the previous presence of endometriosis. Please find some suggestions to improve this review. I would like though to congratulate the authors on their work in reviewing the microRNA part of the manuscript. Furthermore, more elaborated tables and figures are required to summarize the data and increase the citability of this review.
- Please do not abbreviate Endometriosis throughout the text, avoid unnecessary acronyms throughout; usage of excessive acronyms decrease the citability of the paper.
- Line 41-43 = unclear sentence
-
Line 59-60 = unclear sentence
- How about maternal recognition of pregnancy in the tubal epithelium as well as sincitiotrophoblast invasion and extravillous trophoblast remodelling of arteries in ectopic pregnancy? The authors need to be comprehensive and share their views in these processes as well.
-
do endometriotic cells express chemokines that attract macrophages and neutrophis to the lesion, if yes, which, through which mechanisms?
- Line 129-32 = unclear sentence
-
how about fallopian tube infection , fibrosis, adherence and endometriosis/ectopic pregnancy? What is their role in increasing risk of ectopic pregnancy?
-
line 176: abnormal Wnt / ß-catenin pathway ? in which cells of the fallopian tube or in endometriotic cells? This has to be clear throughout
-
The role of hormones is very supperficial, in which cells estradiol acts to promote ectopic pregnancy? How about the role of progesterone, favoring ectopic implantation? Which intracellular pathways are activated by hormones and deregulated? How about growth factors?
- The paragraphs in this review are too long and not very well connected. Efforts to improve flow and continuity of the text is required otherwise the reader will loose interest.
Author Response
Careful consideration was given to the comments of the reviewers, for which we are deeply grateful. Specific responses are provided below.Reviewer: 1Comments to the Author
In this review, Załęcka and collaborators summarized the role of inflammation, WNT pathway and microRNAs in the pathogenesis of endometriosis and ectopic pregnancies. Please find some suggestions to improve this review.
- Please do not abbreviate Endometriosis throughout the text, avoid unnecessary acronyms throughout; usage of excessive acronyms decrease the citability of the paper.
It has been appropriately changed
- Line 41-43 = unclear sentence
It has been appropriately changed
- Line 59-60 = unclear sentence
It has been appropriately changed
- How about maternal recognition of pregnancy in the tubal epithelium as well as sincitiotrophoblast invasion and extravillous trophoblast remodelling of arteries in ectopic pregnancy? The authors need to be comprehensive and share their views in these processes as well.
As suggested, we have added an extra paragraph in the "Potential keypoints" for extravillous trophoblast remodelling of arteries in ectopic pregnancy, which is as follows:
"During the first trimester of pregnancy extravillous trophoblast cells (EVT) invade the maternal decidua. Invasion normally is reduced from the second trimester on-wards and stops in the inner third of the myometrium. In TEP due to specific immuno-logical microenvironments apoptosis induction fails, which deleteriously may result in uncontrolled invasion and penetration of the tubal wall. This process may be probably enhanced by endometriotic lesions through the Slit2/Robo1 signaling pathway".
- do endometriotic cells express chemokines that attract macrophages and neutrophils to the lesion, if yes, which, through which mechanisms?
We supplemented the paragraph on the inflammatory environment with additional information characterizing the recruitment of macrophages and neutrophils to the site of endometriotic lesions. Lines 141-152
- Line 129-32 = unclear sentence
It has been appropriately changed
- how about fallopian tube infection , fibrosis, adherence and endometriosis/ectopic pregnancy? What is their role in increasing risk of ectopic pregnancy?
We presented an additional paragraph of the inflammatory environment section on conflicting signals to the uterine epithelial blastocyst and fallopian tube. We emphasize the role of signals composed of cytokines, chemokines and adhesion molecules that mediate both the adhesion of the blastocyst to the uterine epithelium (and fallopian tubes) and the adhesion of leukocytes to the vascular endothelium. Additionally, we expanded the paragraph to include the potential role of infections in the pathogenesis of EP. Lines 215-234.
We discussed the role of fibrosis in promoting EP in various parts of the review due to its heterogeneous genesis related to both hormonal signaling, chronic inflammation, and the Wnt / ß-catenin pathway. Lines 209-211, 256-261, 314-318.
Additionally, a new figure 2 has been created that summarizes the potential factors in promoting EP.
- line 176: abnormal Wnt / ß-catenin pathway ? in which cells of the fallopian tube or in endometriotic cells? This has to be clear throughout
It has been appropriately changed and supplemented with detailed information on the place of occurrence of the described irregularity.
- The role of hormones is very supperficial, in which cells estradiol acts to promote ectopic pregnancy? How about the role of progesterone, favoring ectopic implantation? Which intracellular pathways are activated by hormones and deregulated? How about growth factors?
Indeed, we agree that the role of hormones has been discussed fairly superficially, so we've expanded the paragraph for additional information. We highlighted the role of estrogen and progesterone and their receptors. In addition, we presented potential interaction pathways involving hormones and their importance in spreading inflammation and, consequently, fibrosis that may contribute to EP. Lines 275-284, 288-301, 314-349.
- The paragraphs in this review are too long and not very well connected. Efforts to improve flow and continuity of the text is required otherwise the reader will loose interest.
We have made every effort to ensure that the paragraphs are not as long as before, while maintaining fluency. However, it was necessary to add a few new paragraphs in response to the reviewers' suggestions.
Reviewer 2 Report
I reviewed the paper entitled “Molecular mechanisms underlying the association between endometriosis and ectopic pregnancy” by Załęcka et al submitted for consideration to IJMS. I found it an interesting topic but I could not really identify the clinical importance of investigating the association of the molecular mechanisms among the two diseases. This is also not clear within the text. More specifically, the clinical impact of the study is not clearly indicated in the text while the authors just present the studies in the literature that may provide common pathways among the two diseases. However, I wonder why we need to find common mechanisms of the two diseases since they are two separate entities and we have no evidence even from case reports or small case studies that the early identification or certain management of the one can prevent the onset of the other. If there is a respective evidence the authors should refer to them so as to bridge the molecular and clinical importance. Additionally, are there limitations to the present study that need to be addressed? If so please provide a respective section. Some other minor things In the introduction section Lines 42-42: The sentence seems a bit unclear. Please clarify. Lines 46- 47: "The main localization..ectopic pregnancy (TEP)-The sentence is incomplete.Author Response
Careful consideration was given to the comments of the reviewers, for which we are deeply grateful. Specific responses are provided below.Reviewer: 2Comments to the Author
I reviewed the paper entitled “Molecular mechanisms underlying the association between endometriosis and ectopic pregnancy” by Załęcka et al submitted for consideration to IJMS.
- I found it an interesting topic but I could not really identify the clinical importance of investigating the association of the molecular mechanisms among the two diseases. This is also not clear within the text.
Endometriosis is a known risk factor for ectopic pregnancy. We have included more literature references to support this thesis, but as rightly mentioned, there is no research so far to prove that early identification of one can prevent the other. However, women with endometriosis are often treated for infertility with Assisted Reproductive Techniques that put them at significant risk of developing EP. Our review may provide a starting point for considering molecular targets in the treatment of both EP and endometriosis. Lines 57-80
- Lines 42-42; Lines 46- 47: unclear sentence
It has been appropriately changed
Reviewer 3 Report
congratulation to the authors for their work on this review article about biological mechanisms occurring in endometriosis and ectopic pregnancy.
this review documents the common mechanisms between the two conditions. It would be interesting to know if the mechanisms mentioned in the text about ectopic pregnancy are of women with pre-existing endometriosis. Could authors add this extra information ? It is not clear in the text.
Also, in the Summary, it would be nice to propose one of the mechanisms - the one that seems to have the most strong association with the two condition - that research could focus right away to possible prevent ectopic pregnancy from occurring in women with endometriosis.
Author Response
Careful consideration was given to the comments of the reviewers, for which we are deeply grateful. Specific responses are provided below.Reviewer: 3Comments to the Author
- It would be interesting to know if the mechanisms mentioned in the text about ectopic pregnancy are of women with pre-existing endometriosis
Unfortunately we do not have such data. The cited literature on ectopic pregnancy did not take into account the endometriosis history of patients.
- in the Summary, it would be nice to propose one of the mechanisms - the one that seems to have the most strong association with the two condition
We highlighted the mechanisms we believe that further research should focus on to possibly prevent ectopic pregnancy in women with endometriosis. Lines 522-525.
Additionally, the interaction paths proposed by us are presented in Figure 2.
Round 2
Reviewer 1 Report
The review improved substantailly, however there are typos and incorrections of english nature that must be correct
Reviewer 2 Report
The authors have clearly addressed and encompassed the requested changes within the manuscript.